# Platycodigenin as Potential Drug Candidate for Alzheimer’s Disease via Modulating Microglial Polarization and Neurite Regeneration

**DOI:** 10.3390/molecules24183207

**Published:** 2019-09-04

**Authors:** Zhiyou Yang, Baiping Liu, Long-en Yang, Cai Zhang

**Affiliations:** 1College of Food Science and Technology, Guangdong Provincial Key Laboratory of Aquatic Product Processing and Safety, Institute of nutrition and marine drugs, Guangdong Ocean University, Zhanjiang 524088, China (B.L.) (L.Y.) (C.Z.); 2Shenzhen Institute of Guangdong Ocean University, Shenzhen 518120, China

**Keywords:** platycodigenin, microglial polarization, PPARγ, neurite regeneration

## Abstract

Neuroinflammatory microenvironment, regulating neurite regrowth and neuronal survival, plays a critical role in Alzheimer’s disease (AD). During neuroinflammation, microglia are activated, inducing the release of inflammatory or anti-inflammatory factors depending on their polarization into classical M1 microglia or alternative M2 phenotype. Therefore, optimizing brain microenvironment by small molecule-targeted microglia polarization and promoting neurite regeneration might be a potential therapeutic strategy for AD. In this study, we found platycodigenin, a naturally occurring triterpenoid, promoted M2 polarization and inhibited M1 polarization in lipopolysaccharide (LPS)-stimulated BV2 and primary microglia. Platycodigenin downregulated pro-inflammatory molecules such as interleukin (IL)-1β, tumor necrosis factor (TNF)-α, IL-6 and nitric oxide (NO), while upregulated anti-inflammatory cytokine IL-10. Further investigation confirmed that platycodigenin inhibited cyclooxygenase-2 (Cox2) positive M1 but increased Ym1/2 positive M2 microglial polarization in primary microglia. In addition, platycodigenin significantly decreased LPS-induced the hyperphosphorylation of mitogen-activated protein kinase (MAPK) p38 and nuclear factor-κB (NF-κB) p65 subunits. Furthermore, the inactivation of peroxisome proliferators-activated receptor γ (PPARγ) induced by LPS was completely ameliorated by platycodigenin. Platycodigenin also promoted neurite regeneration and neuronal survival after Aβ treatment in primary cortical neurons. Taken together, our study for the first time clarified that platycodigenin effectively ameliorated LPS-induced inflammation and Aβ-induced neurite atrophy and neuronal death.

## 1. Introduction

Microglia, the resident brain macrophages, are the main immuno-modulative cells in the central nervous system (CNS) [1,2]. Microglia activation is involved in the pathogenesis of a variety of neurodegenerative diseases including Alzheimer’s disease (AD) [3,4,5]. According to its different roles in immune modulation, microglia can be divided into classically activated M1 and the alternatively activated M2 phenotype [6]. M1 microglia typically release proinflammatory molecules, promote neuroinflammation, and accelerate neuronal death [7]. While activation of M2 microglia increases the secretion of anti-inflammatory factors, resulting in neurogenesis, axonal regeneration, oligodendrogenesis, and anti-neuroinflammation [8]. Lipopolysaccharide (LPS), interferon (IFN)-γ, or Aβ are known as representative M1 microglial polarization stimulators, while interleukin (IL)-4, IL-10, or IL-13 are M2 modulators [9]. A few M1 inhibitive agents, such as cyclooxygenase (Cox)-1 inhibitors, Cox-2 inhibitors have been clinically investigated against neurodegenerative diseases but with little beneficial effects [10]. Thus, only inhibition of M1 microglia is not enough but beneficial M2-skewed microglia activation might also be required for optimizing brain beneficial microenvironment.

Neurite atrophy and synapse loss underlie the pathogenesis of AD and are located upstream of neuronal death in the Aβ cascade [11]. What is more, axonal degeneration probably initiates the onset and progression of neuronal death in AD [12]. Our previous study indicated that neuronal network reconstruction through axonal regeneration was critical for the fundamental memory recovery of AD [13,14]. Studies have shown that inflammatory cytokines activated microglia and induced aggregation of tau in neurites, promoting neuritic dystrophy and neuronal death [15]. Thus, we consider that neurite regeneration is essential for memory reconstruction and stopping secondary neuronal injury via modulating neuroinflammatory microenvironment is required for the maintenance of regenerated neurites in AD.

Peroxisome proliferator-activated receptor (PPAR)-γ, belongs to the nuclear receptor family of ligand-inducible transcription factors, is a known M2 macrophage polarization regulator [16]. PPARγ activation inhibits nuclear factor-κB (NF-κB) signaling pathway via inhibition of IκBα degradation, reduction of RelA (p65) nuclear translocation, and disrupted binding of p65 to the DNA, thus suppresses the gene transcription of inflammatory cytokines [6,17]. The mitogen-activated protein kinase (MAPK) p38 subunit phosphorylation could induce nucleus-export and loss of PPARγ, led to a wide range of inflammatory responses [18,19,20]. In addition, phosphorylation of p38 regulated the association between p65 and p300 via promoting the acetylation of p65, led to increased transcriptional activity of NF-κB [21]. PPAR-γ agonists could attenuate LPS-induced inflammatory response and promote CD206 and Ym1/2 positive M2 microglia polarization [22,23]. However, pharmacological compounds shown to regulate the transformation of microglia to the M2 phenotype as well as promoting neurite regeneration are rare [13,24]. Therefore, drug-targeted M2 microglia polarization and neurite regeneration can provide a new therapeutic strategy for the treatment of AD.

*Platycodon grandifloras* and its main terpenoids platycodigenin-type saponins have been reported for neuroprotective [25,26], anti-tumor, anti-diabetic [27], and anti-inflammatory activities [28,29,30]. However, whether platycodigenin exerts anti-inflammatory effects via modulation of microglial polarization and exact molecular mechanism remains unclear. Moreover, whether platycodigenin could promote neurite regeneration needs to be elucidated. Therefore, in the present study, we investigated the effects of platycodigenin on the proinflammatory mediators’ production in LPS-stimulated BV-2 microglia and explored the underling molecular mechanisms of microglia polarization. In addition, we examined the effects of platycodigenin on neurite regeneration and neuronal survival after Aβ treatment in primary cultured cortical neurons.

## 2. Results

### 2.1. Platycodigenin Inhibited the Production of NO and Ameliorated the Secretion of Pro-Inflammatory Cytokines

To optimize the appropriate concentration of LPS without affecting cell viability, BV2 microglia were treated with LPS (1–1000 ng/mL) for 24 h. The cell viability was not significantly changed compared with control (Appendix A). While the NO release was significantly increased from 10–1000 ng/mL LPS treatment (Appendix A). We chose 100 ng/mL LPS for the following study. To determine the concentration of platycodigenin without cell toxicity, the cells were treated with increasing doses of platycodigenin from 0.01 to 50 μM for 36 h. As a result, the cell viability was not changed after 0.01–10 μM platycodigenin treatment, but decreased to 77.1 ± 1.3% compared with control (Figure 1A). Thus, we chose 10 μM platycodigenin as the maximum concentration in subsequent experiments.

LPS-activated microglia showed a strong release of M1-related NO production, which is one of the main inflammatory mediators and plays an important role in neuro-inflammatory diseases [31]. Thus, we investigated the inhibitory effect of platycodigenin on NO production in LPS-induced BV2 microglia. Compared to control group (4.8 ± 0.1 ng/mL), treatment with LPS significantly increased the production of NO (15.0 ± 0.3 ng/mL). Pretreatment with platycodigenin significantly reduced the NO production in a dose dependent manner at 0.1 (11.6 ± 0.3 ng/mL), 1 (11.3 ± 0.1 ng/mL) and 10 μM (10.7 ± 0.5 ng/mL) (Figure 1B).

In addition, we evaluated the effects of platycodigenin on the LPS-induced production of TNF-α, IL-6, IL-1β, and IL-10, which are important in neuroinflammation and neurodegenerative diseases [4]. Compared to control cells, LPS significantly increased the production of TNF-α, IL-1β, and IL-6 while decreased IL-10 in the culture BV2 supernatants (Figure 1C–E). Pretreatment with platycodigenin for 12 h significantly inhibited TNF-α, IL-1β, and IL-6 production at concentrations of 1 and 10 μM (Figure 1C–E). In addition, platycodigenin ameliorated LPS-induced decrease of IL-10 at 1 μM (Figure 1F).

### 2.2. Platycodigenin Promoted LPS-Induced M1 Phenotype to M2 Microglia Polarization

Cox2, CD11b and inducible nitric oxide synthase (iNOS) are mainly expressed marker proteins in M1 microglia, while Ym1/2, Arginase I, and CD206 are expressed in M2 microglia [32]. We then investigated whether platycodigenin inhibited pro-inflammatory cytokines release via modulating microglial polarization. BV2 microglia were pretreated with platycodigenin (0.1, 1 and 10 μM) for 12 h followed by treatment with LPS (100 ng/mL) for another 12 h (Figure 2A). As a result, platycodigenin significantly inhibited the mRNA expression of TNF-α, IL-1β and iNOS (Figure 2B–D). In the contrary, platycodigenin upregulated M2 microglia related mRNA expression of IL4, CD206, Arg1 and TGFβ (Figure 2E–H). Furthermore, we confirmed the effects of platycodigenin on M2 microglia polarization in primary cultured microglia (Figure 3A). As a result, Cox2 was significantly upregulated and Ym1/2 was downregulated after LPS treatment (Figure 3B,C). Pretreatment with platycodigenin (1 and 10 μM) following LPS treatment significantly decreased Cox2 expression, whereas Ym1/2 expression was significantly increased. The results indicated that platycodigenin exerts anti-inflammatory effects via inhibiting M1 microglial polarization and promoting M2 microglial polarization.

### 2.3. Platycodigenin Attenuated LPS-Induced Activation of MAPK and NF-κB Signaling Pathways

The phosphorylation of p38-MAPK and p65-NF-κB are known to regulate inflammatory mediators and M1 microglial polarization [2,33]. Thereby, we investigated the expression of phosphorylated p38 (*p*-p38) MAPK and phosphorylated p65 (*p*-p65) NF-κB in LPS-induced BV2 microglia. BV2 microglia were treated with platycodigenin (1 and 10 μM) for 12 h following stimulated with LPS for another 1 h, and expression of *p*-p38 and *p*-p65 were analyzed by immunocytochemical analysis (Figure 4A). As a result, p38 and p65 were highly phosphorylated after LPS treatment, while platycodigenin significantly downregulated *p*-p65 and *p*-p38 (Figure 4B–E). To confirm the effects of platycodigenin on p38 and p65 phosphorylation, we measured the expression of *p*-p65 and *p*-p38 by ELISA kits. Phosphorylated p38 and p65 were increased from 1.38 and 20.08 ng/mg to 2.78 and 26.57 ng/mg after LPS treatment, respectively. Platycodigenin treatment significantly downregulated the phosphorylated p65 and p38 (Figure 4F–G).

### 2.4. Platycodigenin as a Potential Agonist for PPARγ

Activation of PPARγ is related to M2 microglial polarization and anti-inflammatory effects [19,23,34]. Therefore, we examined the effect of platycodigenin on PPARγ in BV2 microglia (Figure 5A). We found that LPS treatment significantly suppressed PPARγ expression (Figure 5B), as previously reported [22]. Pretreatment with platycodigenin for 12 h restored PPARγ expression to near normal levels (Figure 5B,C). These results indicated that the anti-inflammatory effect of platycodigenin was possibly through suppression of p38 and p65 signaling and activation of PPARγ.

### 2.5. Platycodigenin Prevents Aβ25-35-Induced Primary Cortical Neuronal Death

To further determine whether platycodigenin showed protective effects on Aβ25-35-induced neuronal death, we treated the platycodigenin to Aβ25-35-induced primary cortical neurons. As shown in Figure 6, compared with control, the neuronal viability was decreased to 62.0% after 3 days Aβ25-35 treatment. Co-treatment with platycodigenin restored Aβ25-35-induced neuronal death to the levels 73.9% (1 μM) and 74.2% (10 μM).

### 2.6. Platycodigenin Promotes Neurites Regeneration after Aβ25-35-Induced Neurites Atrophy

To test the effect of platycodigenin on neurite growth, we investigated its effect on normal cultured cortical neurons (Figure 7A). Compared with control group, PLA dose dependently promoted β3-tubulin positive neurite elongation (Figure 7B,C). Furthermore, we assessed the effects of platycodigenin on Aβ25-35-induced neuritic atrophy in primary neurons (Figure 8A). Neurite density was decreased significantly by Aβ25-35 treatment, whereas treatment with platycodigenin (1 and 10 μM) significantly increased the length of β3-tubulin-positive neurites (Figure 8B,C). These results indicate that platycodigenin can restore Aβ-induced neuritic atrophy and neuronal death.

## 3. Discussion

The hallmarks of Alzheimer’s disease, such as Aβ plaque and neurofibrillary tangles, create toxic microenvironments that trigger activation of M1 microglia phenotype, a process characterized by morphological and phenotype changes [35]. Conversely, activated M1 microglia release inflammatory cytokines which exacerbate CNS microenvironments, preventing neurites remodeling and recovery of memory function [3,4]. Therefore, exploring small molecules that regulate microglial polarization and neurites regeneration might be a therapeutic way for the intervention of AD. In the current study, we demonstrated for the first time that platycodigenin could promote the polarization of LPS-induced M1 microglia to M2 phenotype and enhance neurite regrowth in Aβ-induced axonal atrophy.

Nitric oxide and other reactive nitrogen species have been revealed to induce oxidative stress and enhance expression and DNA binding of NF-κB, promoting neurodegeneration and neuroinflammation [36]. A remarkable increase of M1 associated Cox2 protein leads to overproduction of NO, which are involved in AD [4]. The results of this study suggest that platycodigenin has an inhibitory effect on NO production and attenuates LPS-induced expression of M1 associated Cox2 protein. Pro-inflammatory cytokines, such as TNF-α, IL-1β and IL-6 are remarkably increased, while anti-inflammatory cytokines IL-10, IL4 are decreased in neuroinflammatory and AD models in vitro and in vivo [3]. We found that platycodigenin significantly inhibited the production of the LPS-induced pro-inflammatory cytokines and upregulated anti-inflammatory cytokine in BV2 microglia. NF-κB, which is a critical transcriptional factor in regulating oxidative substances and inflammatory cytokines, plays an important role in inflammatory diseases [5]. The phosphorylation of NF-κB subunit p65 promotes its nuclear translocation and association with CBP/p300 or HDAC-1, increasing the inflammatory gene transcription [6,17]. Furthermore, studies have shown that the transactivation function of NF-κB is dependent on p38 MAPK phosphorylation and p38 mediated p65 acetylation [21]. Thereby, inhibition of p38 and p65 phosphorylation contributes to the reduction of inflammatory response and amelioration of inflammatory diseases. The present study also showed that platycodigenin inhibited LPS-induced phosphorylation of NF-kB p65 and MAPK p38 in BV-2 cells.

PPARγ has been shown to modulate the expression or activity of large numbers of genes in a variety of signaling pathways [34]. Recent animal and clinical studies indicated that PPARγ agonists could prevent inflammatory response and neuronal death in AD [19]. PPARγ agonist not only attenuate the production of pro-inflammatory mediators, such as NO, IL-6, IL-1β, Cox2, iNOS, and TNFα, but also promote the release of anti-inflammatory cytokines, such as IL-10, IL-4, and IL-13. PPARγ agonist pioglitazone inhibits microglia inflammation by blocking p38 MAPK signaling pathways. In addition, PPARγ activation inhibits NF-κB activity by inhibiting nuclear translocation of p65 subunit and competing with NF-κB co-activators [34]. Furthermore, PPARγ activation promotes the expression of M2 markers, such as Ym1/2, Arginase I, CD206, and Fizz1 [23]. Our results showed that platycodigenin increased the expression of PPARγ, probably resulting in the inhibition of *p*-p65 and polarization of M1 microglia to M2 microglia.

Neural circuits and network are responsible for memory generalization, consolidation and storage [37]. Aβ-induced synapse loss, axonal degeneration, and neuronal death are contributed to the disruption of neural networks, which most likely relate to memory deficits [38]. Therefore, we hypothesize the neuronal circuits reconstruction through axonal regeneration is critical for the fundamental therapies of AD. In the current study, by high throughput screening, we found platycodigenin, which could promote neurite regrowth and prevent neuronal death in an Aβ-induced cell model. However, whether platycodigenin could ameliorate memory deficits in vivo needs to be studied further.

In conclusion, we report for the first time that platycodigenin could effectively prevent LPS-induced M1 microglial polarization and inflammatory response by inhibiting *p*-p38 and activating PPARγ, thereby blocking NF-κB activation and reducing the production of oxidative mediators and pro-inflammatory cytokines. Furthermore, platycodigenin prevented Aβ25-35-induced neuronal death and promoted neurite regeneration after Aβ25-35-induced neurite atrophy. Thus, our study exhibited that platycodigenin had the potential for treatment of neuro-inflammation and neuro-degeneration, and can be considered as a promising microglia and neurite modulator for the intervention of AD.

## 4. Materials and Methods

### 4.1. BV2 Microglia and Primary Cortical Microglia Culture

BV2 microglia (The Cell bank of Shanghai Institutes for Biological Sciences, Chinese Academy of Sciences, Shanghai, China) were maintained in Dulbecco’s modified Eagle’s medium (DMEM; Gibco C11995500BT, Rockville, MD, USA) supplemented with 10% fetal bovine serum (FBS; Gibco 10270-106) at 37 °C in a 5% CO_2_/95% air incubator.

Primary microglia were isolated from the cerebral cortices of mice as previously described with minor modifications [24,39]. Briefly, the cerebral cortices were dissected from ICR mice (Guangxi Medical University, Nanning, China) at postnatal day 2–4. After removing dura mater, the tissues were minced, dissociated with 0.25% trypsin-EDTA (Gibco 25200-072) for 30 min at 37 °C, followed by the addition of 600 U/mL DNase I (Gibco 18047-019) and 0.3 mg/mL trypsin inhibitor (Gibco 17075-011) for 15 min. The pellets were collected after centrifugation and suspended in DMEM/F12 (Gibco 12634010) supplemented with 10% FBS. Fifteen percent bovine serum albumin (BSA) solution was added from the bottom of medium to remove debris by centrifugation. Pellets were collected after centrifugation at 740 g for 4 min and re-suspended in 10% FBS-DMEM/F12. Cells were seeded into 6 cm dishes and cultured at 37 °C with 5% CO_2_. After 14–17 days, microglia were isolated from mixed glial cultures with mild trypsinization [39].

### 4.2. Primary Cortical Neuron Culture and Neuronal Viability Assay

Embryos were removed from a pregnant ICR mouse at 14 days of gestation as described previously [13]. For neuronal viability assay, the cells were seeded in a 96-well plate at a density of 20,000 cells per well for 3 d. Then, the neurons were treated with 10 μM Aβ25-35 (Sigma-Aldrich, A4559) or cotreated with platycodigenin (Chengdu Desite Biotech Co. Ltd, Chengdu, China) for 3 days. The Aβ25-35 was previously incubated at 37 °C for 4 days for aggregation. Neurons were then incubated with 10 μL of CCK-8 solution (MCE, HY-K0301) for 3 h. The fluorescent absorbances were measured at 450 nm using an ELISA microplate reader (Biotek, Winooski, VT, USA). For normal neurite density measurement, the cells were seeded in an 8-well plate at a density of 10,000 cells per well for 2 d. Then the neurons were treated with PLA (0.01–10 μM) for 4 d. For Aβ treated neurite density measurement, the neurons were treated with 10 μM Aβ25-35 for 3 days before being treated with platycodigenin for 4 days. The cells were then prepared for immunocytochemistry.

### 4.3. Microglia Viability Assay

MTT assay was performed to detect the effects of LPS and/or platycodigenin on the viability of BV2 microglia. BV2 microglia were seeded in a 96-well plate at a density of 2000 cells per well for 12 or 24 h, then treated with different concentrations of platycodigenin (0.01–50 μM) for another 36 h. Cells were then incubated with MTT (0.5 mg/mL) for 3 h in a 5% CO_2_ incubator at 37 °C. Then the medium was removed and DMSO (150 μL) was added into each well to dissolve formazan crystals. The fluorescent absorbances were measured at 570 nm using an ELISA microplate reader (Biotek, Winooski, VT, USA).

### 4.4. Measurement of Nitric Oxide Levels

BV2 microglia (2 × 10^5^ cells/mL) were seeded in the plates for 12 h, pretreated with platycodigenin for 12 h at different concentrations (0.01, 0.1, 1, 10 μM) followed incubation with LPS (Sigma L2654, 100 ng/mL) for 24 h. Cultured supernatant (100 μL) was collected and added with 100 μL of Griess reagent (1% sulfanilamide and 0.1% naphthylethylenediamine dihydrochloride in 2.5% phosphoric acid; Promega G2930, Madison, WI, USA) for 10 min in the dark at room temperature. An ELISA microplate reader was used for the measurement of absorbances at 540 nm. A standard curve was generated in the same manner using NaNO_2_ for quantitation.

### 4.5. Measurement of TNF-α, IL-6, IL-1β, and IL10

BV2 microglia (2 × 10^5^ cells/mL) were plated in the medium for 12 h, cells were then pretreated with platycodigenin (1 and 10 μM) for 12 h, then 100 ng/mL LPS was added for 24 h. Culture supernatants were collected and enzyme-linked immunosorbent assay (ELISA; dogesce, Beijing, China) was performed for quantification of TNF-α, IL-6, IL-1β, and IL10 according to the manufacturer’s protocol.

### 4.6. RNA Isolation and RT-PCR

BV2 microglia (2 × 10^5^ cells/well) were seeded in 12-well culture plate for 12 h. Cells were then pretreated with platycodigenin (0.1, 1 and 10 μM) for 12 h, then cells were incubated for 12 h in the absence or presence of 100 ng/mL LPS. After incubation, cells were washed twice with cold PBS and total RNA was isolated from the cells using TRI Reagent according to manufacturer’s protocol. Complementary DNA (cDNA) was synthesized from RNA using the GoScript™ Reverse Transcriptase (Promega Corp, Madison, WI, USA). The real-time PCR reaction solution contained a final volume of 25 μl PCR mixture including M-MLV 5X Reaction Buffer 5 μl, dNTP 1.25 μl, recombination RNasin®nuclei acid enzyme inhibitor 25 units, M-MLV reverse transcription enzyme 200 units, adding water to 25 μl. To measure the mRNA level of inflammatory factors and M1/M2 markers including *TNFα* (Forward; 5′-GACGTGGAACTGGCAGAAGAG-3′, Reverse; 5′-TTGGTGGTTTGTGAGTGTGAG-3′), *IL1β* (Forward; 5′-GCAACTGTTCCTGAACTCAACT-3′, Reverse; 5′-ATCTTTTGGGGTCCGTCAACT-3′), *IL4* (Forward; 5′-GGTCTCAACCCCCAGCTAGT-3′, Reverse; 5′-GCCGATGATCTCTCTCAAGTGAT-3′), *iNOS* (Forward; 5′-CTGCAGCACTTGCATCAGGAACCTG-3′, Reverse; 5′-GAGTAGCCTGTGTGCACCTGGAA-3′), *Arg1* (Forward; 5′-TAACCTTGGCTTGCTTCGGAACTC-3′, Reverse; 5′-TGGCGCATTCACAGTCACTTAGG-3′), *CD206* (Forward; 5′-TCTTTGCCTTTCCCAGTCTCC-3′, Reverse; 5′-TGACACCCAGCGGAATTTC-3′), *TGFβ* (Forward; 5′-CTCCCGTGGCTTCTAGTGC-3′, Reverse; 5′-GCCTTAGTTTGGACAGGATCTG-3′), and *Actin* (Forward; 5′-CATCCGTAAAGACCTCTATGCCAAC-3′, Reverse; 5′-ATGGAGCCACCGATCCACA-3′), we designed the primers for target genes (Sangon Biotech, Shanghai, China). The cDNA was amplified using the Supercycler thermal cycler system (Kyratec, Australia) e-Taq DNA polymerase kit (Accurate Biology, Guangzhou, China) and the primers. The PCR products were visualized by CFX96 Touch Real-Time PCR Detection System (Bio-Rad, Life science). Gene expression levels were normalized to the RNA expression of housekeeping gene β-Actin (relative quantification) with the △△CT correction.

### 4.7. Immunocytochemistry

The cells were fixed with 4% paraformaldehyde (Solarbio, Beijing, China) and were blocked with 5% normal goat serum (MultiSciences, Hangzhou, China) in 0.3% Triton-X-PBS and then immunostained for Cox2 (M1 marker, 1:50, Cat. No. sc-514489, Santa Cruz, TX, United States), Ym1/2 (M2 marker, 1:200, Cat. No. ab192029, Abcam, CA, United States), *p*-p65 (1:100, Cat. No. sc-166748, Santa Cruz, TX, United States), *p*-p38 (1:100, Cat. No. sc-166182, Santa Cruz, TX, United States), PPARγ (1:50, Cat. No. sc-7273, Santa Cruz, TX, United States), β3-Tubulin (Neurite marker, 1:200, Cat. No. sc-80005, Santa Cruz, TX, United States), MAP2 (1:2000, Cat No. ab32454, Abcam, CA, United States). Alexa Fluor 488-conjugated goat anti-rabbit IgG, Alexa Fluor 594-conjugated goat anti-mouse IgG (1:200, Cat No. ab150077 and ab150116, Abcam, CA, United States) were used as secondary antibodies. DAPI (Biomol, Hamburg, Germany) was used for counterstaining. Fluorescence images were captured using a fluorescence microscope system (OLYMPUS IX73, Olympus) at size of 259 × 346 μm. Ten slices of each group were captured. The images were analyzed with ImageJ (NIH) as described previously [28].

### 4.8. Measurement of Phosphorylated p65 and p38 by ELISA

BV2 microglia (2 × 10^5^ cells/well) were seeded in 6-well culture plate for 12 h. Cells were then pretreated with platycodigenin (0.1, 1 and 10 μM) for 12 h, then cells were incubated for 1 h in the absence or presence of 100 ng/mL LPS. After incubation, cells were washed twice with cold PBS and total protein was collected from the cells in PBS with cell scraper and sonicated for three times. The total protein was quantified using BCA method and enzyme-linked immunosorbent assay (ELISA; MeiMian, Wuhan, China) was performed for quantification of *p*-p65 and *p*-p38 according to the manufacturer’s protocol.

### 4.9. Statistical Analysis

One-way analysis of variance with post hoc Dunnett’s test and Kruskal-Wallis one-way analysis of variance by ranks test was used for the statistical comparisons by using GraphPad Prism 5 (GraphPad Software, La Jolla, CA, USA). Values of *p* < 0.05 were considered to be statistically significant. The values of the data were expressed as the means ± SEM.

## Figures and Tables

**Figure 1 molecules-24-03207-f001:**
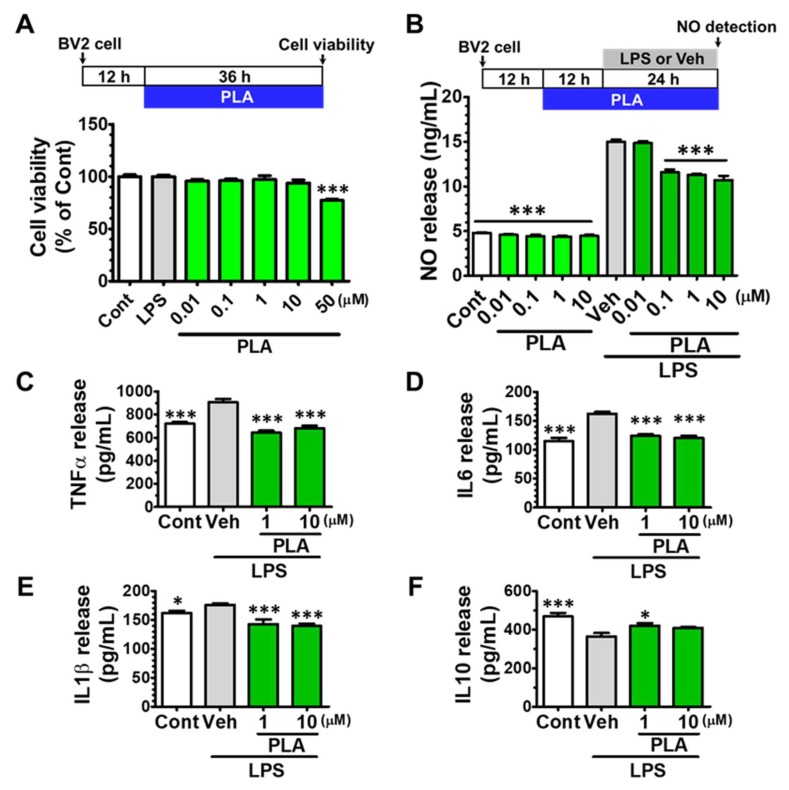
Effects of platycodigenin (PLA) on Lipopolysaccharide (LPS)-induced NO and pro-inflammatory cytokines production in BV2 microglia. BV2 microglia were seeded in 96-well plates for 12 h, followed by treatment with PLA for 36 h (**A**) and the cell viability was assayed. BV2 microglia (2 × 105 cells/mL) were seeded in 48-well plates for 12 h, followed by pretreatment with PLA for 12 h and then LPS (100 ng/mL) was stimulated for another 24 h. NO release (**B**) was detected by Griess reagent method and concentrations of TNF-α (**C**), IL-6 (**D**), IL-1β (**E**) and IL-10 (**F**) were determined by ELISA kit. The data are shown as the mean ± SEM of three independent experiments. Each group has five replicates in the independent experiment. **p* < 0.05, and ****p* < 0.001 versus vehicle (Veh), Kruskal-Wallis one-way analysis of variance by ranks test and one-way analysis of variance and Dunnett’s post hoc test.

**Figure 2 molecules-24-03207-f002:**
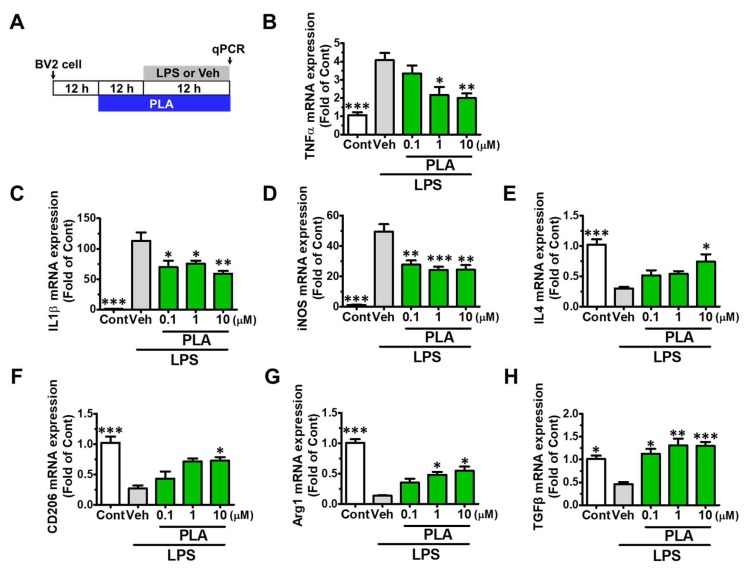
Effects of platycodigenin (PLA) on LPS-induced mRNA expression of M1/M2 microglia markers. BV2 microglia were cultured for 12 h and pretreated with PLA (1 and 10 μM) or control (Cont, 0.1% DMSO) for another 12 h, followed by stimulation with LPS (100 ng/mL) for 12 h. The expression of M1/M2 markers were determined by qPCR analysis. (**A**) Experimental schedule. (**B–H**) The mRNA expression of TNF-α, IL-1β, iNOS, IL4, CD206, Arg1 and TGFβ were quantified. The data are shown as the mean ± SEM of three independent experiments. Each group has four replicates in the independent experiment. **p* < 0.05, ***p* < 0.01, and ****p* < 0.001 versus vehicle (Veh), Kruskal-Wallis one-way analysis of variance by ranks test.

**Figure 3 molecules-24-03207-f003:**
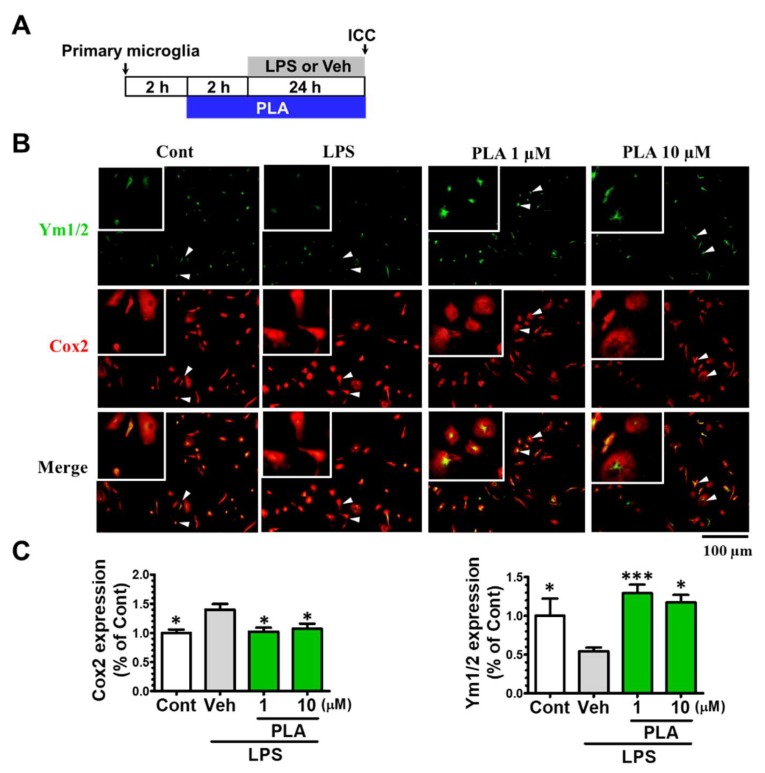
Effects of platycodigenin (PLA) on LPS-induced primary microglial polarization. Primary microglia were cultured for 2 h and pretreated with PLA (1 and 10 μM) or control (Cont, 0.1% DMSO) for another 2 h, followed by stimulation with LPS (100 ng/mL) for 24 h. The expression of Cox2 and Ym1/2 were determined by immunocytochemistry. (**A**) Experimental schedule. (**B**) The representative images immunostained for M1 microglia (Cox2 [red]) and M2 microglia (Ym1/2 [green]). (**C**) Cox2 and Ym1/2 expression per cell were quantified. The data are normalized to the control and shown as the mean ± SEM of three independent experiments. Ten images were captured of each group in the independent experiment. **p* < 0.05 and ****p* < 0.001 versus vehicle (Veh), Kruskal-Wallis one-way analysis of variance by ranks test.

**Figure 4 molecules-24-03207-f004:**
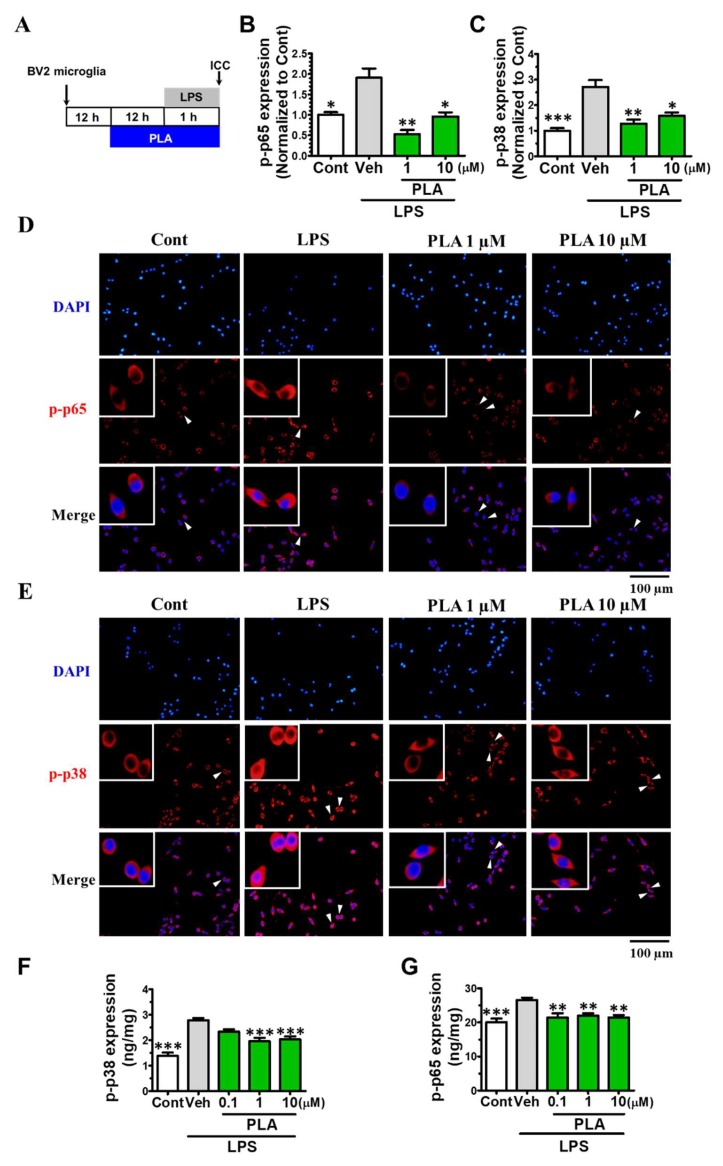
Effects of platycodigenin (PLA) on LPS-induced phosphorylation of NF-κB p65 and MAPK p38. BV2 microglia were cultured for 12 h and pretreated with PLA (1 and 10 μM) or control (Cont, 0.1% DMSO) for another 12 h, followed by stimulation with LPS (100 ng/mL) for 1 h. The cells were fixed and immunostained for *p*-p65 and *p*-p38. The fluorescent intensity was measured. (**A**) Experimental schedule. Phosphorylation of p65 (**B**) and p38 (**C**) per cell were quantified. The representative images of *p*-p65 (**D**) and *p*-p38 (**E**) were showed. (**F,G**) Phosphorylation of p65 and p38 per mg total protein were quantified by ELISA kit. The data are normalized to the control and shown as the mean ± SEM of three independent experiments. Ten images were captured (B-E) and four replicates (F-G) were performed of each group in the independent experiment. **p* < 0.05, ***p* < 0.01, and ****p* < 0.001 versus vehicle (Veh), Kruskal-Wallis one-way analysis of variance by ranks test and one-way analysis of variance and Dunnett’s post hoc test.

**Figure 5 molecules-24-03207-f005:**
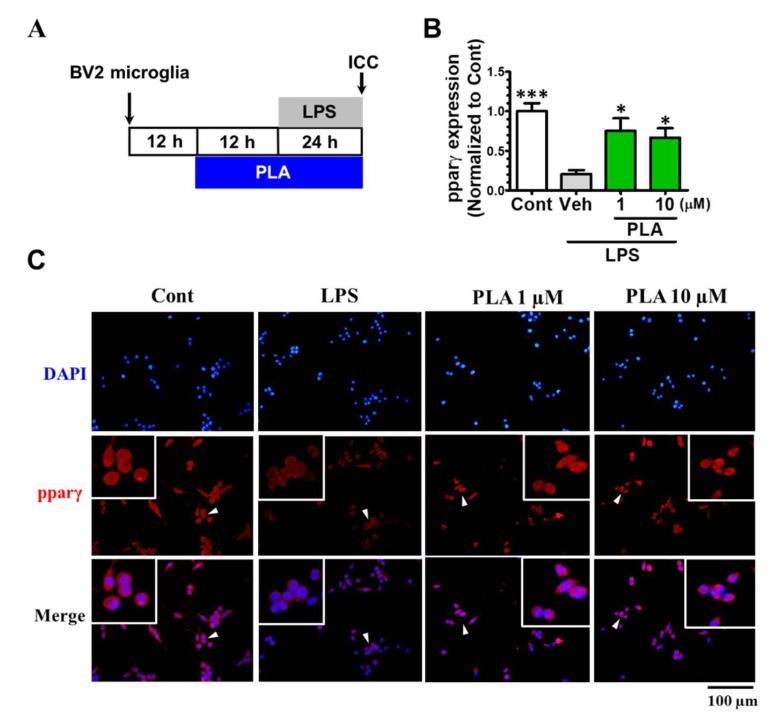
Effects of platycodigenin (PLA) on LPS-induced expression of PPARγ. BV2 microglia were cultured for 12 h and pretreated with PLA (1 and 10 μM) or control (Cont, 0.1% DMSO) for another 12 h, followed by stimulation with LPS (100 ng/mL) for 24 h. The cells were fixed and immunostained for pparγ. The fluorescent intensity of the pparγ positive cells were measured. (**A**) Experimental schedule. (**B**) The expression of PPARγ per cell were quantified. (**C**) The representative images immunostained for PPARγ. The data are normalized to the control and shown as the mean ± SEM of three independent experiments. Ten images were captured of each group in the independent experiment. **p* < 0.05 and ****p* < 0.001 versus vehicle (Veh), Kruskal-Wallis one-way analysis of variance by ranks test.

**Figure 6 molecules-24-03207-f006:**
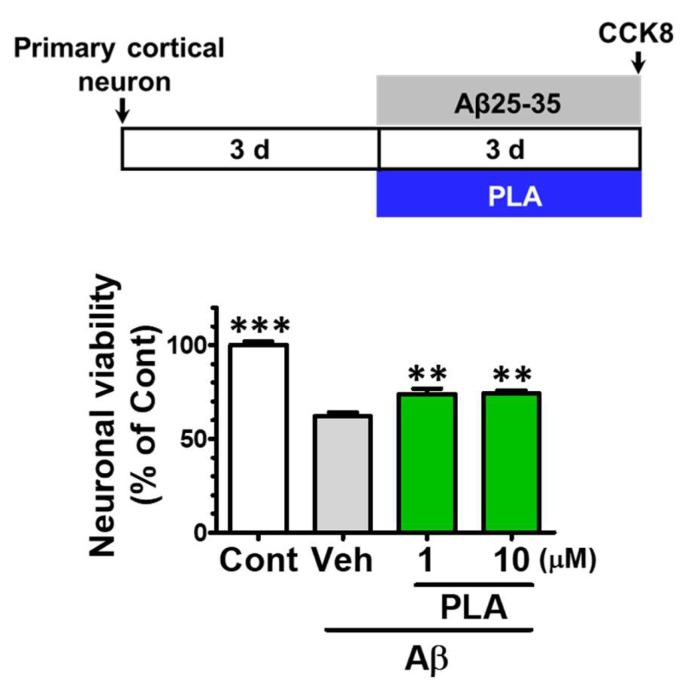
Effects of platycodigenin (PLA) on Aβ25-35-induced neuronal death. Primary cultured cortical neurons were cultured for 3 d and treated with PLA (1 and 10 μM) and/or Aβ25-35 (10 μM) for another 3 d. The neuronal viability was determined by CCK8 analysis. The data are normalized to the control and shown as the mean ± SEM of three independent experiments. Each group has five replicates in the independent experiment. ***p* < 0.01 and ****p* < 0.001 versus vehicle (Veh), Kruskal-Wallis one-way analysis of variance by ranks test.

**Figure 7 molecules-24-03207-f007:**
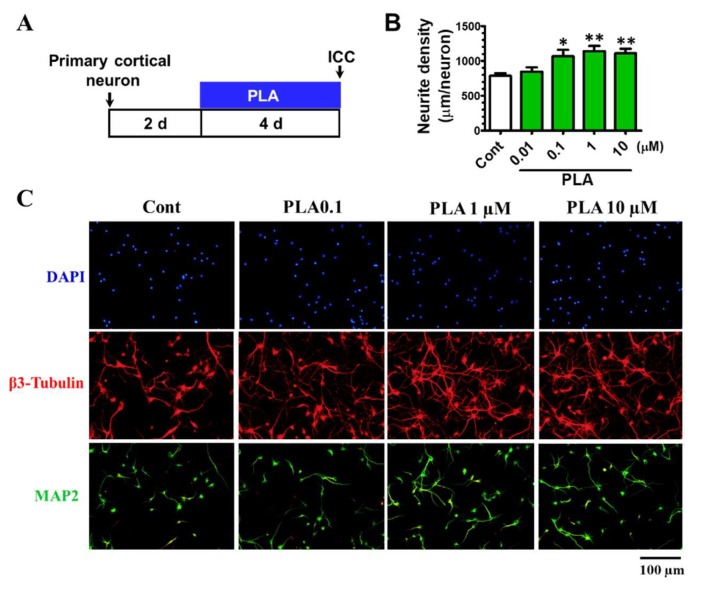
Effects of platycodigenin (PLA) on neurite outgrowth. Primary cultured cortical neurons were cultured for 2 d following treated with PLA (0.01–10 μM) or control (Cont, 0.1% DMSO) for another 4 d. The neurons were fixed and immunostained for β3-tubulin and MAP2. The lengths of the β3-tubulin positive neurites were measured. (**A**) Time schedule of the experiments. (**B**) Quantification of total neurite outgrowth and associated statistics. (**C**) Representative images of β3-tubulin and MAP2 positive neurites. Scale bar, 100 μm. The data are shown as the mean ± SEM of three independent experiments. Ten images were captured of each group in the independent experiment. **p* < 0.05 and ***p* < 0.01 versus control (Cont), one-way analysis of variance and Dunnett’s post hoc test.

**Figure 8 molecules-24-03207-f008:**
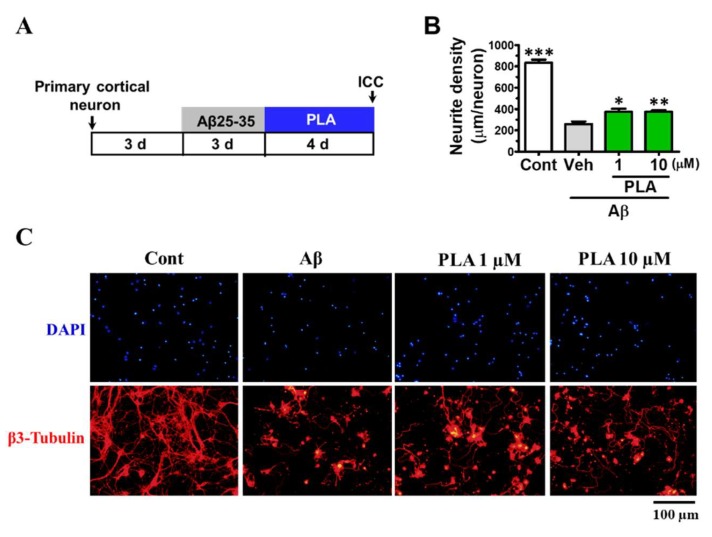
Effects of platycodigenin (PLA) on Aβ25-35-induced neuritic atrophy. Primary cultured cortical neurons were cultured for 3 d and pre-treated with Aβ25-35 (10 μM) for 3 d, then PLA (1 and 10 μM) or control (Cont, 0.1% DMSO) were treated to neurons for another 4 d. The neurons were fixed and immunostained for β3-tubulin. The lengths of the β3-tubulin positive neurites were measured. (**A**) Time course of the experiments. (**B**) Quantification of total neurite outgrowth and associated statistics. (**C**) Representative images of β3-tubulin-positive neurites in Aβ25-35 treated group. Scale bar, 100 μm. The data are shown as the mean ± SEM of three independent experiments. Ten images were captured of each group in the independent experiment. **p* < 0.05, ***p* < 0.01 and ****p* < 0.001 versus vehicle (Veh), one-way analysis of variance and Dunnett’s post hoc test.

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
