# Peer review of "Platycodigenin as Potential Drug Candidate for Alzheimer’s Disease via Modulating Microglial Polarization and Neurite Regeneration"

_molecules, 2019, doi:10.3390/molecules24183207_

Round 1

Reviewer 1 Report

Compounds derived from natural sources are gaining increasing interest as novel medicines and leads for treating several highly prevalent diseases for which there are still no efficient prevention or treatment available. This is even more crucial for low grade inflammatory diseases related with aging as is the case of neurodegenerative disorders such as Alzheimer’s disease (AD) that are increasing in our aging society. The identification of potential effective and safe bioactive compounds and the knowledge on their mechanisms of action is of main importance for its future use in the clinical practice. Based on this, the current manuscript is addressing an important and very actual thematic.  In this manuscript the authors are testing the hypothesis of whether the saponin platycodigenin was an effective drug to target M2 polarization using mouse microglia cell line (BV2) /primary cells and neurite regeneration using a primary culture of mouse cortical neurons. The authors provide a clear introduction on the thematic and the main objectives are well described. The manuscript is well written and clear.

In terms of the cell culture experimental design, the results are described to correspond to the analyses of triplicate experiments. The question is which was the value of n (the number of independently performed cell experiments)? Replicates are not independent tests and for that reason, this type of experiments cannot provide evidence of reproducibility of the results. The independent experiments must have n≥3 so that error bars and statistics can be shown.  The authors should revise this concept and clearly state the number of independent experiments performed.

The immunocytochemistry  results are shown as representative images of one experiment and the images are clear. Nevertheless, this is the only methodology used by the authors to sustain the results on protein expression and regulation and their conclusions.

For confirmation and strengthening of these results, different and complementary studies must be also performed. Namely the study of gene/protein expression of selected M1/M2 markers by qPCR/western blot or even cell cytometry, should be added. Also, to conclude on the mechanism of action and regulation of the MAPK and NF-κB signalling pathways confirmation at the level of gene or protein expression is needed. In this case it will be necessary to perform the experiments with statistical significance using n≥3 independent experiments.

In terms of methodologies and results, although in each figure, the included experiment schematics are clear and help clarify the experiments methods, some important details in the manuscript  (and included in the figures/results) are not yet very clear and consistent in terms of description (control experiment and/or vehicle) and needs clarification and homogenization thorough the manuscript. As an example the authors state:

In 4.2. Primary cortical neuron culture and neuronal viability assay, Line 277-279, the authors describe the vehicle experiment as treatment with (0.1% DMSO) in alternative to platycodigenin. Then in Line 126-128 (Figure 2) authors  describe as experimental controlor solvent control (Cont, 0.1% DMSO)”. In line 95 the authors also refer to control experiment as untreated cells. The use and nature of control experiment (cont) and vehicle (Veh) needs clarification.

Author Response

Response to Reviewer 1 Comments

Point 1: In terms of the cell culture experimental design, the results are described to correspond to the analyses of triplicate experiments. The question is which was the value of n (the number of independently performed cell experiments)? Replicates are not independent tests and for that reason, this type of experiments cannot provide evidence of reproducibility of the results. The independent experiments must have n≥3 so that error bars and statistics can be shown.  The authors should revise this concept and clearly state the number of independent experiments performed.

Response 1: Thank you for your comments, we performed over three independent experiments for each experiment, we are sorry that we made a mistake for the words and expression, and we corrected them in L113, L137, L147, L170, L188, L200, L217, L227.

Point 2: The immunocytochemistry results are shown as representative images of one experiment and the images are clear. Nevertheless, this is the only methodology used by the authors to sustain the results on protein expression and regulation and their conclusions.

For confirmation and strengthening of these results, different and complementary studies must be also performed. Namely the study of gene/protein expression of selected M1/M2 markers by qPCR/western blot or even cell cytometry, should be added. Also, to conclude on the mechanism of action and regulation of the MAPK and NF-κB signalling pathways confirmation at the level of gene or protein expression is needed. In this case it will be necessary to perform the experiments with statistical significance using n≥3 independent experiments.

Response 2: Thanks for the suggestions, we performed additional experiments (qPCR) to confirm the effects of platycodigenin on inflammatory response and M1/M2 microglia polarization (TNFα, IL1β, iNOS, IL4, CD206, Arg1, TGFβ) in L120-124, L132-138, L336- L361.

In addition, to evaluate the effects of platycodigenin on the regulation of the MAPK and NF-κB signalling pathways, we performed PathScan® ELISA experiments to confirm the expression of pp38 and pp65 in BV2 cells in L157-161, L168-169, L375-381.

Point 3: In terms of methodologies and results, although in each figure, the included experiment schematics are clear and help clarify the experiments methods, some important details in the manuscript  (and included in the figures/results) are not yet very clear and consistent in terms of description (control experiment and/or vehicle) and needs clarification and homogenization thorough the manuscript. As an example the authors state:

In 4.2. Primary cortical neuron culture and neuronal viability assay, Line 277-279, the authors describe the vehicle experiment as treatment with (0.1% DMSO) in alternative to platycodigenin. Then in Line 126-128 (Figure 2) authors  describe as experimental control “or solvent control (Cont, 0.1% DMSO)”. In line 95 the authors also refer to control experiment as untreated cells. The use and nature of control experiment (cont) and vehicle (Veh) needs clarification.

Response 3: Thanks for your useful comments, we have corrected the confused expression thorough the manuscript. L95, L133, L142, L165, L183, L212, L223.

Reviewer 2 Report

This study investigated the possible therapeutic effect of platycodigenin in treating AD through modulating microglial polarization and neurite regeneration. Although this finding is impressive, the experimental designs were not appropriate that hamper the acceptance of this manuscript. The drawback of this manuscript is that they use different cells to interpret the same issue, For instance, they use BV2 cells to measure pro-inflammatory cytokines production, phosphorylation of NF-κB p65 and MAPK p38, and expression of PPARγ and primary microglial cell to study polarization. It is strongly recommended that they study the same issue in one system, instead of using two systems and then stitching the results together as a whole story.

The other minor questions were listed below.

The ICC in Figure 3 is missing. The use of One-way analysis of variance is incorrect. This method is used in the parametric analysis. Since the values were normalized and nonparametric statistics were required. Thus, Kruskal-Wallis one-way analysis of variance by ranks was appropriate.

Author Response

Response to Reviewer 2 Comments

Point 1: This study investigated the possible therapeutic effect of platycodigenin in treating AD through modulating microglial polarization and neurite regeneration. Although this finding is impressive, the experimental designs were not appropriate that hamper the acceptance of this manuscript. The drawback of this manuscript is that they use different cells to interpret the same issue, For instance, they use BV2 cells to measure pro-inflammatory cytokines production, phosphorylation of NF-κB p65 and MAPK p38, and expression of PPARγ and primary microglial cell to study polarization. It is strongly recommended that they study the same issue in one system, instead of using two systems and then stitching the results together as a whole story.

Response 1: Thank you for your useful suggestions, we have performed additional qPCR experiments to support our finding that platycodigenin regulates M1/M2 microglia polarization using BV2 cells in L120-124, L132-138, L336-L361. Furthermore, we confirmed the effects of platycodigenin on M2 microglia polarization in primary cultured microglia.

Point 2: The other minor questions were listed below.

The ICC in Figure 3 is missing. The use of One-way analysis of variance is incorrect. This method is used in the parametric analysis. Since the values were normalized and nonparametric statistics were required. Thus, Kruskal-Wallis one-way analysis of variance by ranks was appropriate.

Response 2: We have added the ICC photos in Figure 4D and 4E in L163-172. Thank you for the suggestion of statistical analysis, we have corrected the statistical method using Kruskal-Wallis one-way analysis of variance by ranks in L114, L138, L147, L171, L188, L201.

Reviewer 3 Report

The manuscript claimed Platycodigenin as potential durg candidate for AD through microglia modulation and neurite regeneration. The results is premature and need to be furture polished.

Major comments: 

The authors should add the contration 0.1 uM of PLA treatment in all function assay as there seem to be the effect of PLA is not dose dependent. The authors should put fluorescence imaging in the figure 3 main figure and also use westernblot to double confirm the results. Figure 6 cannot make the conclusion that PLA can promote neurites regeneration which lack of control(is PLAtreament on neuron can induce neurite outgrowth?)

Author Response

Response to Reviewer 3 Comments

Point 1: The manuscript claimed Platycodigenin as potential durg candidate for AD through microglia modulation and neurite regeneration. The results is premature and need to be furture polished.

Major comments:

The authors should add the contration 0.1 uM of PLA treatment in all function assay as there seem to be the effect of PLA is not dose dependent. The authors should put fluorescence imaging in the figure 3 main figure and also use westernblot to double confirm the results. Figure 6 cannot make the conclusion that PLA can promote neurites regeneration which lack of control(is PLAtreament on neuron can induce neurite outgrowth?)

Response 1: Thank you for your suggestions. We made corrections as below.

We performed additional qPCR experiments to evaluate the effect of PLA from 0.1 to 10 uM on inflammatory response and M1/M2 microglia polarization in BV2 cells. L120-124, L132-138, L336-L361.

We have put representative fluorescence image in the figure 4, in addition, to confirm our results, we performed ELISA to evaluate the expression of pp38 and pp65 in figure 4. L157-161, L168-169, L375-381.

In our previous experiment, we had found that PLA promoted neurite outgrowth in normal cultured neurons, we added the result in figure 7. P203-205, P211-218.

Round 2

Reviewer 1 Report

The authors have tried to answer most of the comments. Comment 2 and 3 are fully addressed although still dependent of comment 1 that was not yet fully answered. In the modifications the authors performed regarding comment 1 (significance of cell culture experiments results)  it`s still not clear the  experimental design that was performed to generate the data used in the statistics. The authors  describe to have performed three independent cell experiments, but do not describe the number of replicates performed in each independent cell experiment.  To be clear,  independent cell experiments means that the cell experiments have to be repeated in different times and from a different stock cell culture. If we are describing different cell wells in one plate performed at the same time in the same conditions, this would mean replicates and do not reflect the reproducibility of the difference between the experimental cells and the control cells, just the accuracy of pipetting (replicate wells). In order to have a complete experimental description to support the significance of the results presented, this detailed description needs to be included in the manuscript. This is important to infer the significance of the results.

Author Response

Response to Reviewer 1 Comments

Point 1: The authors have tried to answer most of the comments. Comment 2 and 3 are fully addressed although still dependent of comment 1 that was not yet fully answered. In the modifications the authors performed regarding comment 1 (significance of cell culture experiments results)  it`s still not clear the  experimental design that was performed to generate the data used in the statistics. The authors  describe to have performed three independent cell experiments, but do not describe the number of replicates performed in each independent cell experiment.  To be clear,  independent cell experiments means that the cell experiments have to be repeated in different times and from a different stock cell culture. If we are describing different cell wells in one plate performed at the same time in the same conditions, this would mean replicates and do not reflect the reproducibility of the difference between the experimental cells and the control cells, just the accuracy of pipetting (replicate wells). In order to have a complete experimental description to support the significance of the results presented, this detailed description needs to be included in the manuscript. This is important to infer the significance of the results.

Response 1: Thank you so much for your sincere and invaluable comments, we also agree with you that the detailed experimental description and reproducibility of the data is very important. We added those information in the manuscript and also as below.

In figure 1, for the cell viability, NO release and inflammatory cytokines detection, three stocked BV2 cells were used in each independent experiment, and each group of them has five replicates (repeated wells).

In figure 2, for the qPCR analysis, three stocked BV2 cells were used in each independent experiment, they were performed at the same time, and each group of them has four replicates (repeated wells).

In figure 3, for the Cox2 and Ym1/2 immunostaining, we performed three independent experiment, and ten photos of each group were captured in each experiment.

In figure 4, for the NF-κB p65 and MAPK p38 immunostaining, we performed three independent experiment, and ten photos of each group were captured in each experiment. For the ELISA, three stocked BV2 cells were used in each independent experiment, they were performed at the same time, and each group of them has four replicates (repeated wells).

In figure 5, for the PPARγ immunostaining, we performed three independent experiment, and ten photos of each group were captured in each experiment.

In figure 6, for the effects of platycodigenin (PLA) on Aβ25-35-induced neuronal death, we performed primary cortical neuronal culture for 3 times, each group has five replicates (repeated wells) in the independent experiment.

In figure 7 and 8, for the effects of PLA on neurite outgrowth, we performed primary cortical neuronal culture for 3 times, and ten photos of each group were captured in each experiment.

Reviewer 2 Report

Although the author re-presented mRNA data form BV2 cells to amend the integrity of this manuscript, mRNA did not always correlate with the expressed protein.

Author Response

Response to Reviewer 2 Comments

Point 1: Although the author re-presented mRNA data form BV2 cells to amend the integrity of this manuscript, mRNA did not always correlate with the expressed protein.

Response 1: Thank you for your useful suggestions, we appreciate your comment. Although it’s best for us to perform WB or ICC staining to check M1 and M2 expressed markers using BV2 cells, our ELISA data in figure 1, qPCR data in figure 2 and ICC result in figure 3 could indirectly or directly supply evidences that PLA promote M2 microglia and inhibit M1 microglia polarization. Furthermore, we think that our data in primary cultured microglia are more convinced.

Again, we thank you for your comments and we will perform the complete experiment in one system and confirm our results in another cell system in our future study.

Reviewer 3 Report

The authors have answered all my questions. I suggest this manuscript published in molecules.

Author Response

Thank you for your recognition of our efforts.